

# Identification of immunological and prognostic value of SHROOM2 in pan-cancer and experimental verification of its role in promoting malignant phenotypes in breast cancer

Yaya Wang[1], Yuechao Ren[2], Xiaoyan Zheng[1], Yan Wang[1], Haoqi Wang[3], Xi Zhang[3] and Sainan Li[3]

[1] School of Agroforestry and Medicine, The Open University of China, BeiJing, China
[2] Research and Development Center, The Open University of China online Education & Information Technology Co., Ltd, BeiJing, China
[3] Department of Breast Center, The Fourth Hospital of Hebei Medical University, Shijia Zhuang, Hebei, China

Corresponding author
Sainan Li, 48402144@hebmu.edu.cn

## ABSTRACT

**Objective**. This study aimed to investigate the relationship between Shroom Family Member 2 (SHROOM2) expression and immune features, survival outcomes, and tumor mutational burden (TMB) across various cancer types, as well as its impact on the aggressive behavior of breast cancer (BC).

**Methods**. RNA sequencing and clinical survival data were retrieved from the TCGA-BRCA and TCGA-PANCANCER datasets within The Cancer Genome Atlas (TCGA) database. Survival analyses were performed to assess the association between SHROOM2 expression and clinical outcomes across different cancer types. Gene set enrichment analysis (GSEA) was applied to identify potential mechanisms associated with differentially expressed genes in BC. Spearman correlation and Wilcoxon tests were used to evaluate the relationships between SHROOM2 levels and immune characteristics, TMB, or drug sensitivity. Additionally, the effects of SHROOM2 on BC cells were assessed through reverse transcription-quantitative polymerase chain reaction (PCR), cell counting kit-8 (CCK-8) assay, transwell assay, and wound healing assay.

**Results**. SHROOM2 was overexpressed in a range of cancers, and its upregulation was associated with poor clinical outcomes. It was significantly correlated with immunomodulators, immune checkpoints, and immune cell infiltration in pan-cancer. Moreover, it showed an association with TMB and drug sensitivity in BC. Increased SHROOM2 expression enhanced the proliferative, migratory, and invasive capabilities of BC cells *in vitro*.

**Conclusion**. High SHROOM2 expression is linked to poor prognosis in BC, supporting its potential as both an immunological and predictive biomarker.

## INTRODUCTION

Cancer ranks among the foremost causes of global mortality, presenting significant challenges to both human health and social well-being. It is estimated that one in five individuals, regardless of gender, will develop cancer over the course of their lifetime, with nearly one in nine men and one in twelve women succumbing to cancer-related fatalities (*Bray et al., 2024*). Data from the International Agency for Research on Cancer (IARC) indicate that, in 2022, there were approximately 20 million new cancer cases and 9.7 million cancer-related deaths worldwide (*Bray et al., 2024*). Projections for 2025 forecast 2,041,910 new cancer diagnoses and 618,120 cancer deaths in the United States alone (*Siegel et al., 2025*). Consequently, efforts to reduce cancer incidence and enhance patient prognosis remain central to ongoing research initiatives.

Advances in cancer biology have revolutionized cancer management, shifting from single-modality treatments, such as surgery and endocrine therapy, to multimodal approaches combining surgery, radiotherapy, and chemotherapy (*Sonkin, Thomas & Teicher, 2024*). Medical technology breakthroughs have exposed the limitations of tumor cell-targeted therapies, including therapeutic resistance and the development of aggressive subclones. As a result, immunotherapy has become an established clinical strategy targeting the tumor immune microenvironment (TIME), thereby enhancing immune cell-mediated tumor cytotoxicity and overcoming these challenges (*Liu & Dilger, 2025*). Although these approaches have improved patient prognosis and reduced incidence, tumor heterogeneity continues to drive treatment failure and resistance in certain patient subsets. Biomarkers are therefore indispensable in predicting therapeutic efficacy and optimizing clinical outcomes. Ongoing biomarker discovery remains vital for refining treatment strategies, uncovering new targets, and ultimately improving patient survival (*Joshi et al., 2024*).

Shroom Family Member 2 (SHROOM2) is implicated in ocular albinism type 1 (OA1) due to its localization on the X chromosome and is recognized as a human homolog of apical protein in *Xenopus* (APX) (*Hagens et al., 2006*). Several studies have linked SHROOM2 to the risk and pathogenesis of esophageal squamous carcinoma, colorectal cancer, and medulloblastoma (*Closa et al., 2014*; *Liu et al., 2024*; *Shou et al., 2015*). SHROOM2 expression is significantly reduced in nasopharyngeal carcinoma (NPC) cells compared to normal nasopharyngeal epithelial cells. Loss of SHROOM2 enhances NPC cell migration, invasion, and metastatic potential, while also promoting epithelial-to-mesenchymal transition (EMT) (*Yuan et al., 2019*), thereby facilitating NPC progression. Furthermore, SHROOM2 expression has been associated with cellular proliferation, apoptosis, and migration in hepatocellular carcinoma (*Chen et al., 2025*). Analysis *via* the Genecard database reveals a strong association with the VEGFA-VEGFR2 signaling pathway, a crucial regulator of angiogenesis within tumor-associated vasculature, which plays a central role in the TIME. However, the role of SHROOM2 in other cancer types, particularly breast cancer (BC), remains unclear, warranting further investigation.

The quantification of non-synonymous somatic mutations in the cancer cell genome defines tumor mutational burden (TMB), which varies across different cancer types and within the same category. Early retrospective and prospective studies identified

TMB as a potential predictor of immune checkpoint inhibitor response (*Budczies et al., 2024*). This leads to the approval of pembrolizumab for patients with high TMB levels, as evidenced by data from the Keynote-158 trial (*Maio et al., 2022*). While ongoing research seeks to determine its universal applicability across various malignancies and optimal TMB thresholds, investigations are advancing along three primary avenues: first, refining TMB assessment through rigorous quality control measures to address limitations such as restricted assay ranges and low tumor purity; second, improving conventional TMB models by incorporating factors like clonality, persistence, human leukocyte antigen (HLA)-adjusted TMB, tumor neoantigen load, and mutation profiles; and third, integrating TMB with both existing and emerging biomarkers, including programmed cell death ligand-1 (PD-L1) expression, microsatellite instability, immune-related gene signatures, and the tumor immune microenvironment (*Kiri & Ryba, 2024*; *Trocchia et al., 2024*). A comprehensive understanding of the mechanisms underlying TMB is essential due to its crucial role in cancer progression and its impact on the immune system's ability to recognize tumors. The relationship between SHROOM2 and TMB remains unexplored, but investigating this connection could reveal key factors that drive tumor progression.

Recent advancements in transcriptomics have incorporated innovative methodologies, including generative adversarial networks (GANs) (*Ai, Smith & Feltus, 2023*). This study utilized transcriptomic data from The Cancer Genome Atlas (TCGA) to assess SHROOM2 expression across various cancers, investigating its correlations with prognosis, clinicopathological features, TMB, and immune characteristics in pan-cancer, with a specific focus on BC. Additionally, *in vitro* experiments were performed to confirm SHROOM2's functional involvement in BC pathogenesis. This study expands the understanding of SHROOM2's expression profile across cancers and its relationship with tumor immunity, identifying SHROOM2 as a potential oncogene and an immune infiltration-related biomarker, particularly in BC.

# MATERIALS & METHODS

## Differential expression analysis

RNA-sequencing (RNA-seq) data across multiple cancer types were retrieved from TCGA (https://portal.gdc.cancer.gov) and the Genotype-Tissue Expression (GTEx, http://www.gtexportal.org/) databases using the STAR pipeline. The data were processed and converted to the transcripts per million (TPM) format. To ensure data quality, duplicate and incomplete RNA-seq samples were excluded, and the remaining data were transformed to the log2(TPM + 1) format. SHROOM2 expression levels were assessed between normal and tumor tissues *via* the Wilcoxon rank-sum test, with specimens stratified into high- and low-expression groups based on the median SHROOM2 expression threshold.

## Survival and clinicopathological analyses

Clinical follow-up and clinicopathological data were sourced from TCGA (https://portal.gdc.cancer.gov). To reduce potential bias in statistical analysis, individuals with incomplete overall survival (OS) data, those with a follow-up duration of less than 30 days, and male patients with BC were excluded from both the breast cancer gene (BRCA)

and pan-cancer datasets. Progression-free interval (PFI) was defined as the time from the initiation of treatment to cancer progression or death from any cause. Disease-specific survival (DSS) was calculated as the time from diagnosis to death attributable to the specific cancer, with deaths from other causes treated as censored. OS was defined as the time from diagnosis to death from any cause. Survival analysis was conducted using Cox proportional hazards regression, with statistical significance determined at $P < 0.05$.

Samples from TCGA-BRCA were categorized based on estrogen receptor (ER) status, progesterone receptor (PR) status, age, and PAM class. SHROOM2 expression was evaluated and compared across these categories using the Wilcoxon rank-sum test. Additionally, survival analysis was performed within subgroups stratified by these clinicopathological characteristics to assess the prognostic value of SHROOM2 expression in BC.

### Enrichment analysis

Gene set enrichment analysis (GSEA) was performed to assess the potential impact of SHROOM2 levels on cancer-associated pathways, with the results displayed in a heatmap. Furthermore, the role of SHROOM2 expression in BC was explored by applying GSEA to identify pathways associated with genes influenced by SHROOM2 in BC.

### Assessment of immune characteristics

SHROOM2's impact on immune-related features was evaluated by examining its association with immunomodulators, immune checkpoints, and immune cell infiltration (ICI). Gene catalogs for immune checkpoints and immunomodulatory factors were retrieved from TISIDB (http://cis.hku.hk/TISIDB/download.php). Immunomodulatory scores were then compared between high- and low-SHROOM2-expression groups to determine the relationship between SHROOM2 expression and immunomodulatory genes across cancer types.

ICI data were sourced from TIMER2.0 (http://timer.cistrome.org/), and infiltration scores for various immune cell populations were assessed using the TIMER, EPIC, MCP-COUNTER, CIBERSORT, CIBERSORT-ABS, QUANTISEQ, and XCELL tools. Heatmaps were generated to visualize the correlation between ICI and SHROOM2 expression levels across cancers. Furthermore, the ESTIMATE algorithm was employed to analyze these associations in both pan-cancer and BC.

Immune activity scores were calculated using the Tumor Immunophenotype tool (http://biocc.hrbmu.edu.cn/TIP/). These scores were compared between high- and low-SHROOM2-expression groups to evaluate the link between SHROOM2 expression and immune activity.

Finally, immune checkpoint scores were compared between the high- and low-SHROOM2-expression cohorts to assess the correlation between SHROOM2 expression and immune checkpoint genes across cancer types.

### TMB and drug sensitivity analyses

TMB and drug sensitivity analyses were conducted to assess tumor heterogeneity and predict the efficacy of immunotherapy. TMB scores in the pan-cancer dataset were calculated using

the "maftools" package (*Mayakonda et al., 2018*). The relationship between *SHROOM2* expression and TMB scores was then examined.

The chemotherapeutic response of each sample was predicted using drug sensitivity data from the Genomics of Drug Sensitivity in Cancer (GDSC) database (https://www.cancerrxgene.org/), the largest publicly available pharmacogenomic resource (*Geeleher, Cox & Huang, 2014*). The "pRRophetic" R package was employed for prediction, with half-maximal inhibitory concentration ($IC_{50}$) values determined *via* ridge regression (*Jiang et al., 2021*). Default settings were used for all parameters. Batch effects were corrected using ComBat, and gene expression values from different tissues were averaged to address duplicates.

## Cell culture and siRNA transfection

To explore the involvement of SHROOM2 in breast cancer pathogenesis, *in vitro* experiments were performed using MCF7 (ATCC HTB-22) and MDA-MB-231 (ATCC HTB-26) cell lines. Both cell lines were cultured in DMEM supplemented with 10% FBS at 37 °C in a 5% $CO_2$ atmosphere. SHROOM2-specific siRNA duplexes (GenePharma, custom-designed) and negative control siRNA (NC) were transfected into the cells using the siRNA-Mate plus reagent (#G04026; GenePharma), following the optimized protocol outlined in Table S1. Knockdown efficiency was assessed 48 h post-transfection *via* qRT-PCR. Each biological experiment included three independent replicates.

## CCK-8 assay

To assess the role of SHROOM2 in breast cancer *in vitro*, SHROOM2 expression was knocked down using siRNA, and the subsequent effects on cellular behavior were evaluated. The siRNA sequence used was: forward, GCACAUCUGAGCAGUUCUATT; reverse, UAGAACUGCUCAGAUGUGCTT. Cell proliferation was measured by CCK-8 assay. Specifically, MCF7 and MDA-MB-231 cells were seeded into 96-well plates at a density of 5,000 cells/100 µL of culture medium. The effect of SHROOM2 knockdown on cell proliferation was determined by measuring absorbance at 450 nm at 0 h, 24 h, 48 h, and 72 h intervals.

## Transwell assay

Cell migration was assessed using a Transwell assay. MCF7 and MDA-MB-231 cells were resuspended in DMEM without fetal bovine serum (FBS) at a concentration of $2 \times 10^4$ cells/mL. A 100 µL aliquot of the cell suspension was added to the upper compartment of a Transwell insert, while the lower chamber contained 500 µL of DMEM supplemented with FBS. After the incubation period, non-migratory cells were removed with a cotton swab, and cells that had migrated through the membrane were fixed with 4% formaldehyde, stained with 0.1% crystal violet, and counted.

To assess invasive capacity, a mixture of DMEM without FBS was prepared at a 1:7 ratio with stroma (Beyotime, Guangzhou, China). A 50 µL volume of this mixture was added to the upper chamber, and incubation was carried out for 3–4 h. The subsequent steps were identical to those used in the migration assay.
## qRT-PCR

Total RNA was extracted from MCF7 and MDA-MB-231 cells using the Vazyme RNA Isolation Kit (Vazyme, Nan Jing, China). Complementary DNA (cDNA) synthesis was performed from the isolated RNA following the manufacturer's protocol with a reverse transcription kit (Vazyme). The resulting cDNA was amplified using the following primers: forward, AGTTCTACTCGCGCTTCTGT; reverse, CCTTCATGTAGCTGAGCCCT. Gene expression levels were quantified using the $2^{-\Delta\Delta CT}$ method based on the Ct values obtained *via* quantitative PCR.

## Wound healing assay

Cells were seeded into 6-well plates at a density of $1 \times 10^6$ cells per well. After transfection and overnight incubation, a linear wound was created across the confluent monolayer using a 10-$\mu$L micropipette tip. Images were captured at the start of the experiment (0 h) and after 24 h under 5$\times$ magnification.

## Statistical analysis

Statistical comparisons between two groups were conducted using Student's $t$-test. Differences among multiple groups were evaluated with the Wilcoxon rank-sum and Kruskal-Wallis tests. Cancer outcomes were analyzed using the log-rank test and Cox regression analysis. Statistical significance was defined as $P < 0.05$.

# RESULTS

## SHROOM2 was highly expressed in pan-cancer

Analysis of RNA-seq data from TCGA and GTEx repositories revealed significant overexpression of SHROOM2 in cancers such as adenoid cystic carcinoma (ACC), bladder urothelian carcinoma (BLCA), cervical squamous cell carcinoma and endocervical adenocarcinoma (CESC), colon adenocarcinoma (COAD), colon adenocarcinoma/rectum adenocarcinoma (COADREAD), esophageal carcinoma (ESCA), gliobastoma multiforme (GBM), lower-grade glimas (GBMLGG), kidney chromophobe (KICH), acute myeloid leukemia (LAML), brain lower grade glioma (LGG), liver hepatocellular carcinoma (LIHC), lung adenocarcinoma (LUAD), lung squamous cell carcinoma (LUSC), ovarian carcinoma (OV), pancreatic adenocarcinoma (PAAD), prostate adenocarcinoma (PRAD), skin cutaneous melanoma (SKCM), stomach adenocarcinoma (STAD), stomach and esophageal carcinoma (STES), testicular germ cell tumors (TGCT), thyroid carcinoma (THCA), and uterine carcinosarcoma (UCS) (Figs. 1A–1B). In BC, SHROOM2 expression was significantly higher in tumor samples compared to healthy breast tissue (Fig. 1C).

## SHROOM2 was correlated with prognosis in pan-cancer

SHROOM2 levels correlated with OS in BRCA, kidney renal clear cell carcinoma (KIRC), and rectum adenocarcinoma (READ) (Fig. 2A); DSS in KIRC and sarcoma (SARC) (Fig. 2B); and PFI in KIRC, PAAD, SARC, THCA, and uveal melanoma (UVM) (Fig. 2C). Kaplan–Meier curves were generated to illustrate the relationship between SHROOM2 levels and prognosis in different cancers (Figs. 2D–2M).

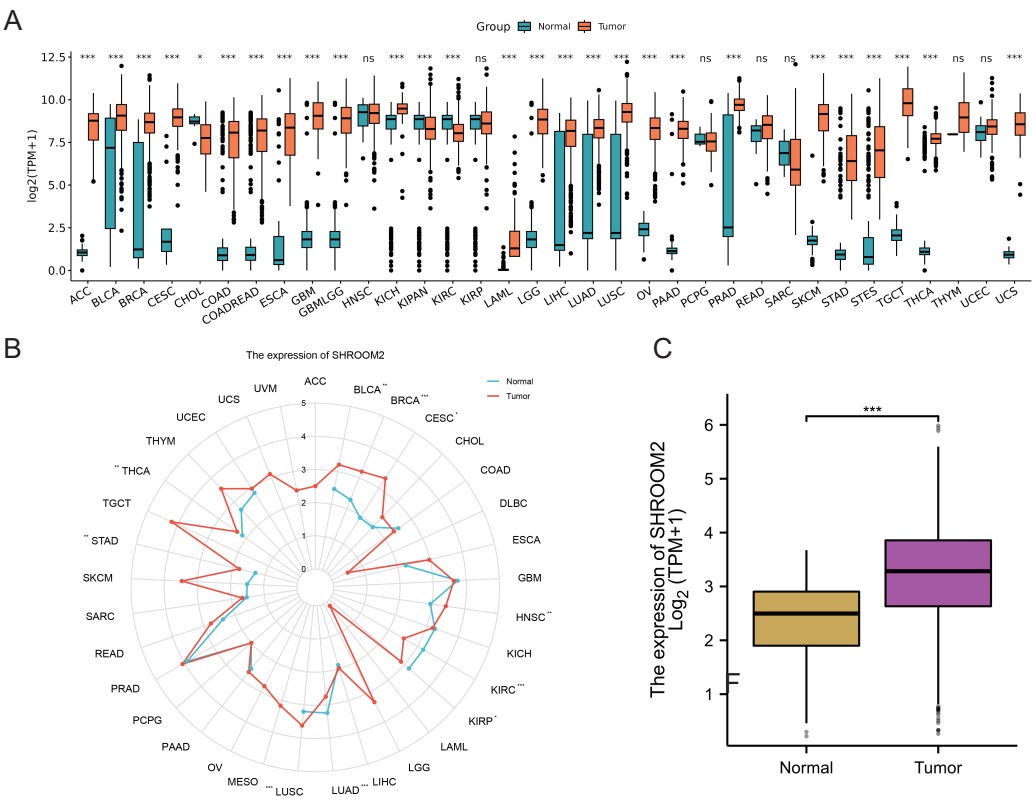

**Figure 1** **Expression of SHROOM2 in pan-cancer (including breast cancer).** (A) *SHROOM2* expression in pan-cancer based on microarray data from TCGA and GTEx databases. (B) *SHROOM2* expression in pan-cancer based on microarray data from TCGA database. (C) *SHROOM2* expression in breast cancer and normal breast tissues in TCGA-BRCA dataset.

Additionally, SHROOM2 expression was associated with several clinicopathological features. Higher levels were observed in patients aged ≤60 years (Fig. 3A), those with positive estrogen receptor (ER) status (Fig. 3B), progesterone receptor (PR) status (Fig. 3C), and luminal A and luminal B subtypes (Fig. 3D). SHROOM2 also exhibited high diagnostic accuracy, with an area under the curve (AUC) of 0.775 (Fig. 3E).

Subgroup analysis was conducted to assess the impact of SHROOM2 on survival outcomes in BC patients with varying clinicopathological features (Figs. 3F–3O). The results revealed a significant association between SHROOM2 expression and poorer survival in ER-positive (Fig. 3H), PR-positive (Fig. 3J), and luminal A BC (Fig. 3L) groups, suggesting a strong correlation between these traits and SHROOM2 levels.

## SHROOM2 was correlated with multiple cancer-related signaling pathways

Pathway analysis was performed to explore the role of SHROOM2 in cancer-related mechanisms. Pan-cancer GSEA identified SHROOM2 enrichment in several cancer-associated pathways, including EMT (Fig. 4A). In BC, GSEA demonstrated SHROOM2's association with DNA methylation, N-cadherin, and estrogen signaling pathways (Fig. 4B).

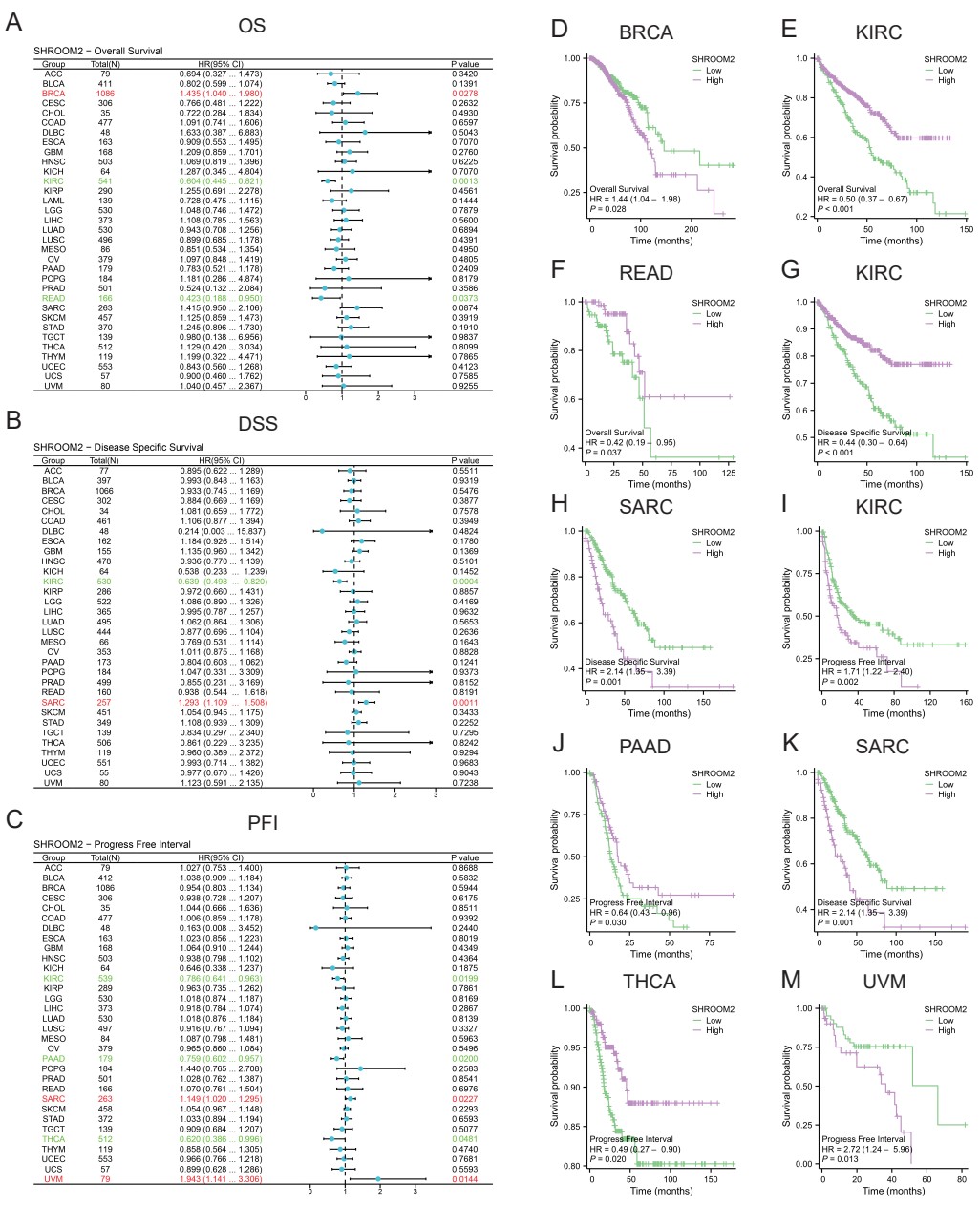

**Figure 2** **Relationship between *SHROOM2* expression and prognosis in pan-cancer.** (A) Relationship between *SHROOM2* expression and OS in pan-cancer (univariate Cox analysis). (B) Relationship between *SHROOM2* expression and DSS in pan-cancer (univariate Cox analysis). (C) Relationship between *SHROOM2* expression and PFI in pan-cancer (univariate Cox analysis). Prognostic value of *SHROOM2* expression in terms of OS in (D) BRCA, (E) KIRC, (F) and READ; DSS in (G) KIRC and (H) SARC; and PFI in (I) KIRC, (J) PAAD, (K) SARC, (L) THCA and (M) UVM. OS, overall survival; DSS, disease-specific survival; PFI, progression-free interval; BRCA, breast invasive carcinoma; KIRC, kidney renal clear cell carcinoma; READ, rectal adenocarcinoma; SARC, sarcoma; THCA, thyroid carcinoma; UVM, uveal melanoma.

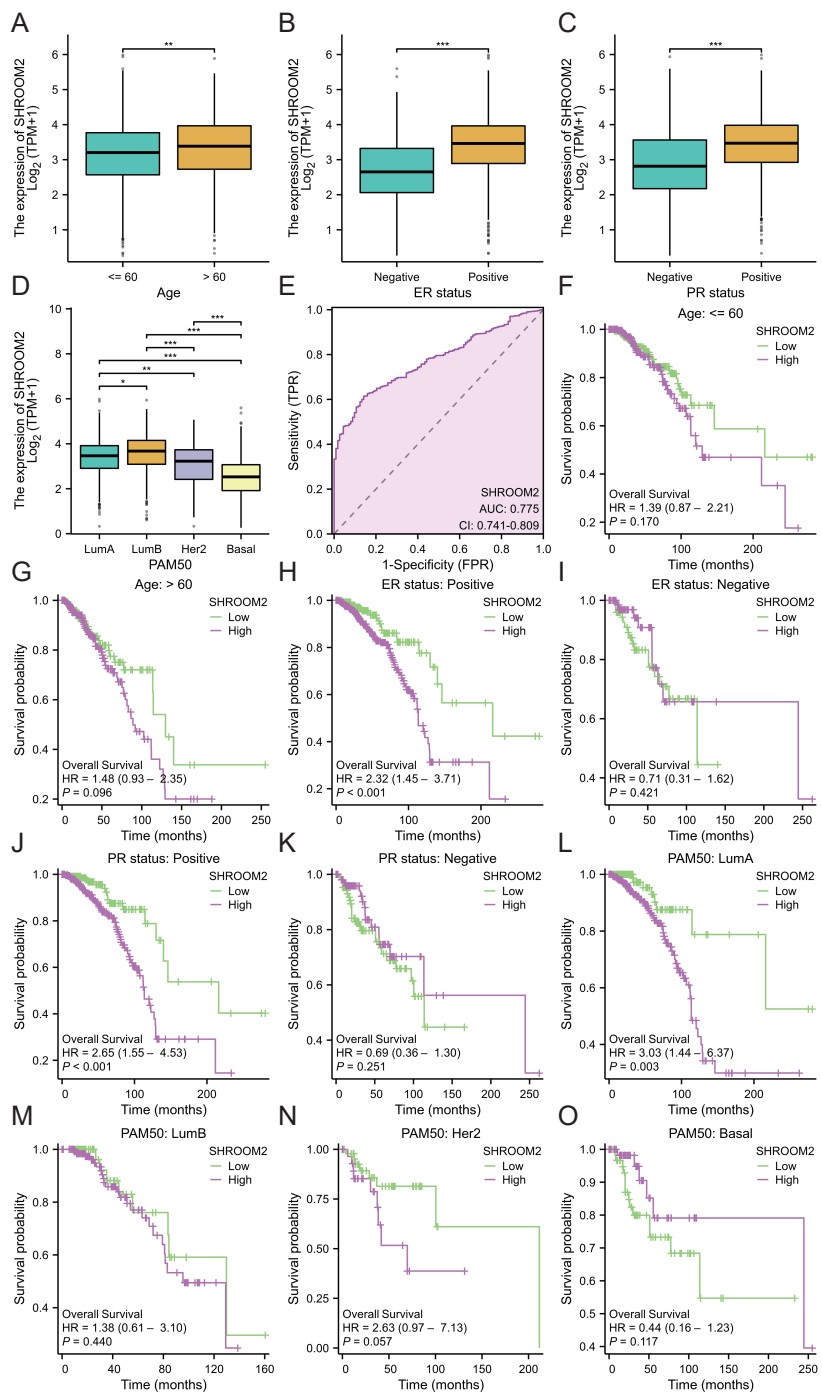

**Figure 3  Relationship between SHROOM2 expression and clinicopathological characteristics.** SHROOM2 expression in different groups stratified based on (A) age, (B) ER status, (C) PR status, and (D) PAM50 subtypes. (E) Receiver operating characteristic (ROC) curve demonstrating the diagnostic efficacy of SHROOM2 in breast cancer. Subgroup KM analysis based on (F–G) age, (H–I) ER status, (J–K) PR status, and (L–O) PAM50 subtypes. ER, estrogen receptor; PR, progesterone receptor; PAM50, Prediction Analysis of Microarray 50 gene panel; KM, Kaplan–Meier.

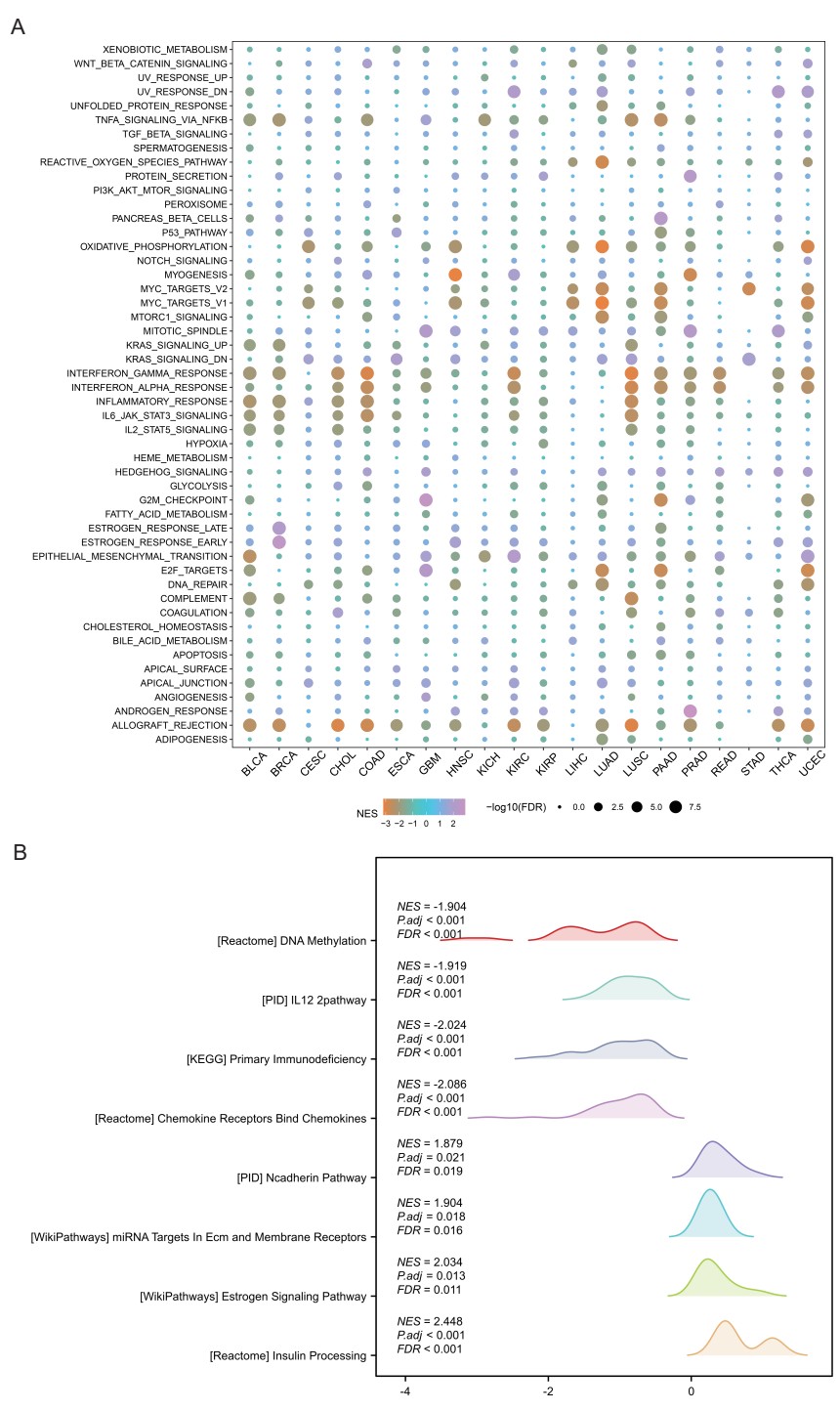

**Figure 4** **Enrichment analysis of upregulated and downregulated genes in two groups stratified based on the median *SHROOM2* expression.** (A) Pan-cancer GSEA. (B) GSEA analysis based on the *SHROOM2* median expression grouping in breast cancer. GSEA , Gene Set Enrichment Analysis; DEGs, differentially expressed genes.

## SHROOM2 affected immunomodulators, immune checkpoints, ICI, immune activity, and TMB in pan-cancer

Spearman correlation analysis revealed a strong association between SHROOM2 expression and that of immunomodulators in pan-cancer (Fig. 5).

Additionally, SHROOM2 levels were found to correlate with immune checkpoint expression across various cancer types (Fig. 6A). Notably, SHROOM2 expression was positively correlated with PD-L1 (CD274) in BRCA, CESC, COAD, COADREAD, GBM, KICH, kidney renal papillary cell carcinoma (KIRP), LGG, LIHC, LUSC, PRAD, SARC, STES, TGCT, thymoma (THYM), and Wilms tumor (WT) (Fig. 6A).

As illustrated in Fig. 6B, SHROOM2 expression was strongly associated with inflammation-related pathways in multiple cancer types.

Furthermore, as shown in Fig. 7, SHROOM2 levels were linked to the presence of various immune cell types across pan-cancer. In BC, SHROOM2 levels were correlated with immune, stromal, and ESTIMATE scores (Fig. 8A). The distribution of immune cell types in BC samples is presented in Fig. 8B. Notably, SHROOM2 expression was associated with the infiltration of B cells, CD4+ T cells, CD8+ T cells, cancer-associated fibroblasts, macrophages, neutrophils, and mast cells in BC (Fig. 8C). In addition, immune activity scores were significantly higher in the low-SHROOM2-expression cohort compared to the high-SHROOM2-expression cohort (Fig. 8D).

Analysis of the relationship between SHROOM2 expression and TMB (Figs. 9A–9K) indicated a positive correlation between SHROOM2 levels and TMB in LAML (Fig. 9F) and THYM (Fig. 9J), while a negative correlation was observed in COAD (Fig. 9A), ESCA (Fig. 9B), HNSC (Fig. 9C), KIRC (Fig. 9D), KIRP (Fig. 9E), LIHC (Fig. 9G), LUAD (Fig. 9H), STAD (Fig. 9I), and UCEC (Fig. 9K).

## SHROOM2 expression was associated with drug sensitivity in BC

The relationship between SHROOM2 expression and clinical response to anti-cancer treatments was assessed by examining its correlation with various common anti-cancer agents (Figs. 10A–10H). The data revealed that increasing SHROOM2 expression was associated with elevated $IC_{50}$ values for standard chemotherapeutic agents, including docetaxel (Fig. 10A), paclitaxel (Fig. 10B), doxorubicin (Fig. 10C), cisplatin (Fig. 10D), 5-Fluorouracil (Fig. 10E), and methotrexate (Fig. 10F) (Figs. 10A–10F). In contrast, SHROOM2 expression did not significantly influence the efficacy of targeted or endocrine therapies (Figs. 10G–10H). These results suggest that SHROOM2 expression modulates the sensitivity of BC cells to chemotherapeutic agents.

## SHROOM2 induced breast cancer cell proliferation, migration, and invasion *in vitro*

*In vitro* experiments were conducted to evaluate the impact of SHROOM2 expression on the biological behavior of BC cells. As demonstrated in Figs. 11A–11B, SHROOM2 expression was successfully silenced in MCF7 and MDA-MB-231 BC cell lines *via* siRNA transfection. CCK-8 assays showed that SHROOM2 knockdown significantly reduced the proliferation rate of both cell lines (Figs. 11C–11D).

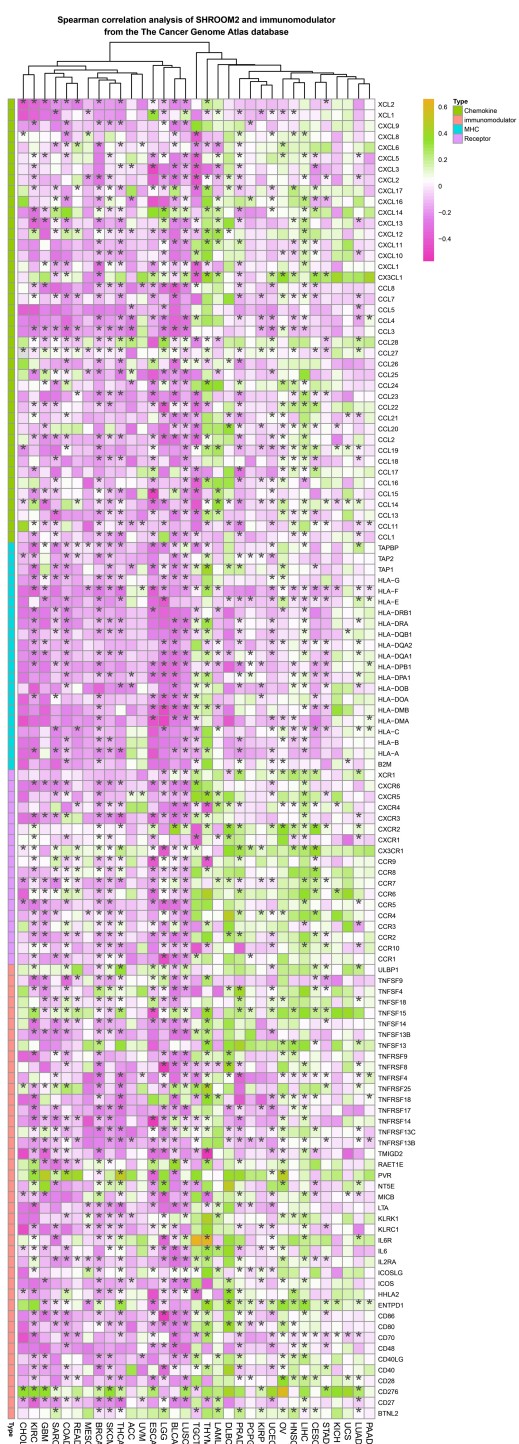

**Figure 5  Correlation between *SHROOM2* expression and immunomodulators in pan-cancer.**

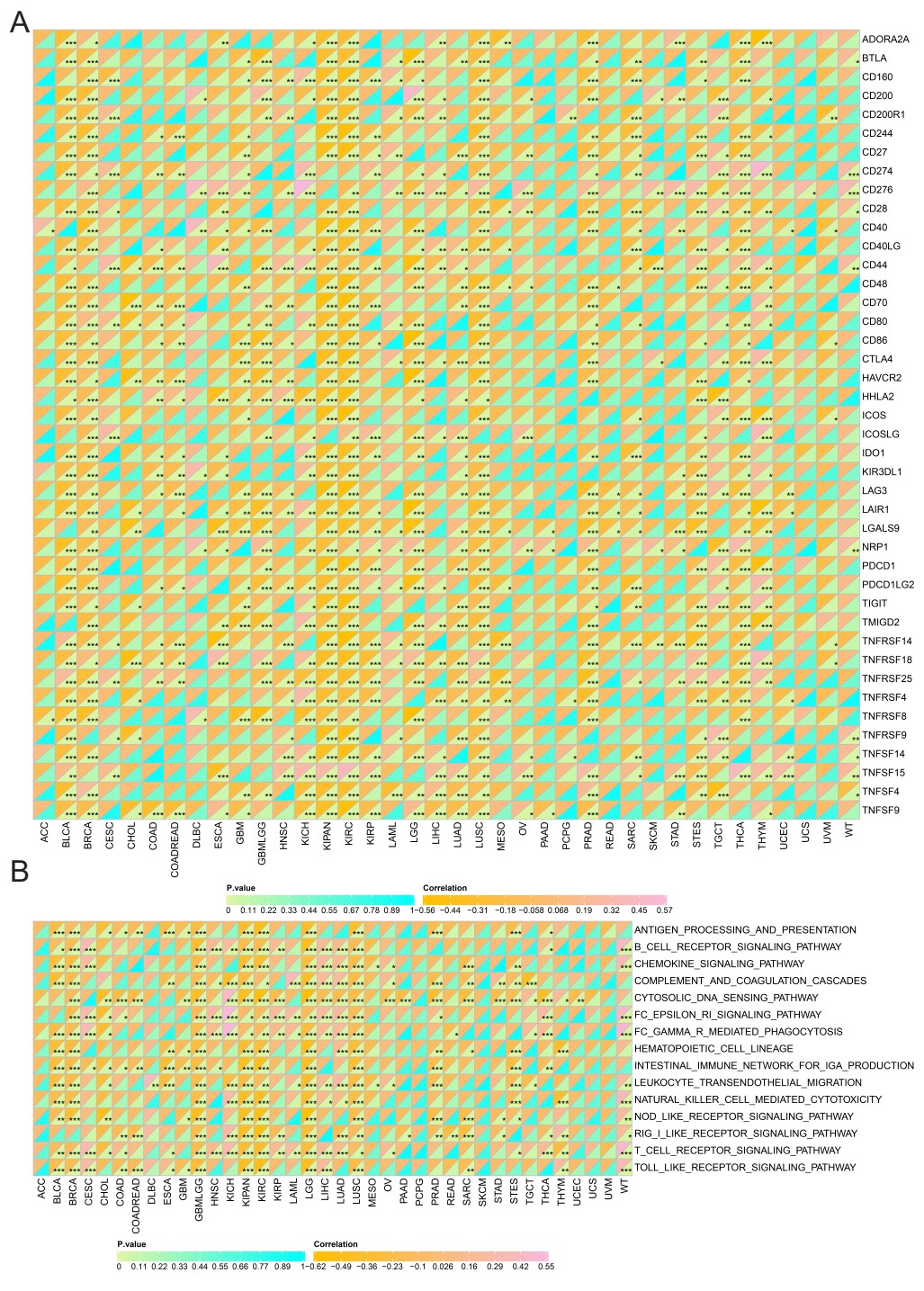

**Figure 6** **Correlation of *SHROOM2* expression with immune checkpoints, and inflammation in pan-cancer.** (A) Correlation between *SHROOM2* expression and immune checkpoints in pan-cancer. (B) Correlation between *SHROOM2* expression and inflammation in pan-cancer. "ns" represents not significant. *, $P < 0.05$; **, $P < 0.01$; ***, $P < 0.001$.

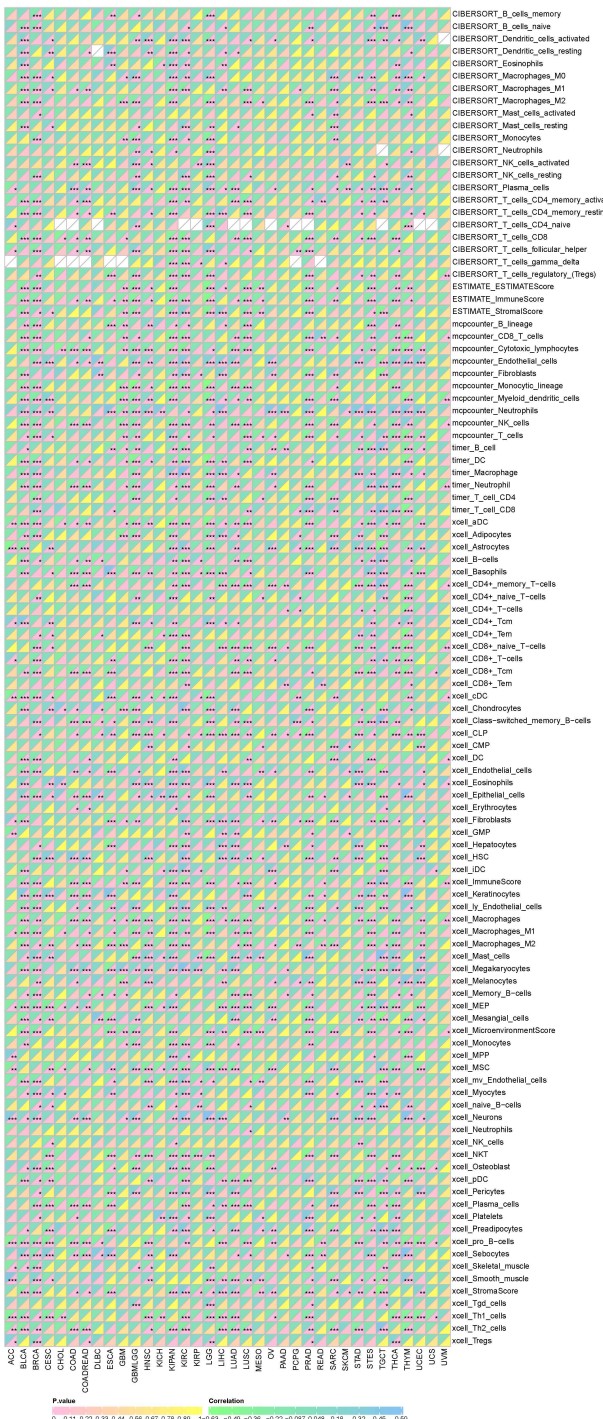

**Figure 7** Correlation between *SHROOM2* expression and immune cell infiltration in pan-cancer.

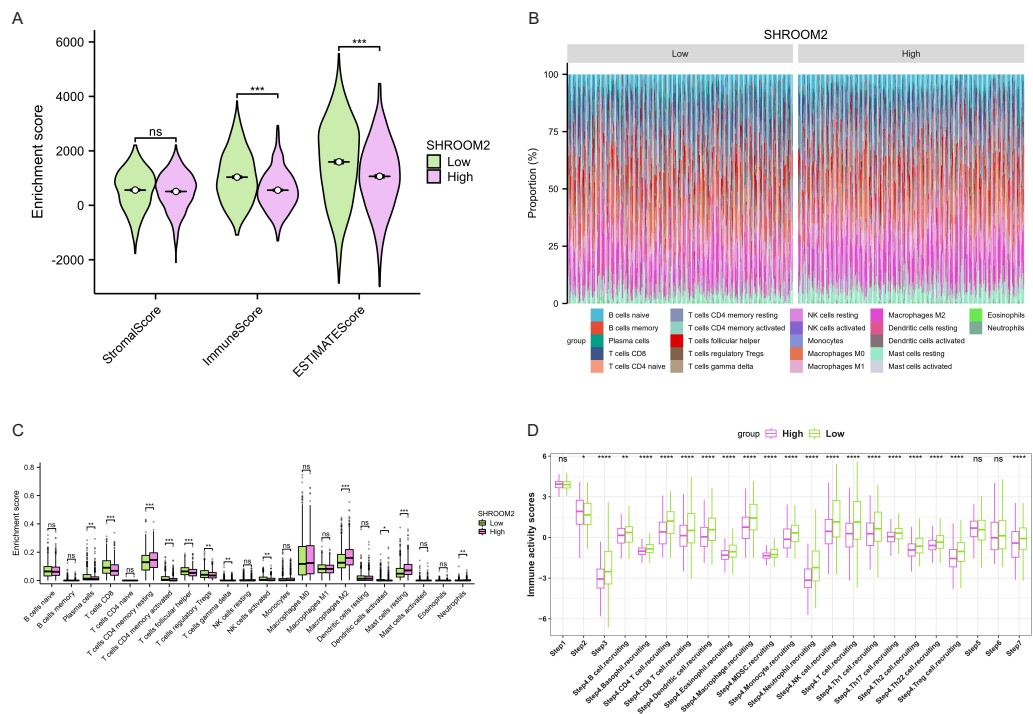

**Figure 8** **Correlation between *SHROOM2* expression and immune cell infiltration and immune activity in breast cancer.** (A) Correlation between *SHROOM2* expression and the ESTIMATE, immune, and stromal scores in breast cancer. (B) Immune cell infiltration levels in different breast cancer samples (analyzed using the CIBERSORT algorithm). (C) Relationship between *SHROOM2* expression and immune cell infiltration in breast cancer (analyzed using the CIBERSORT algorithm). (D) Relationship between *SHROOM2* expression and immune activity score in breast cancer. STEP1: Release of cancer cell antigens; STEP2: Cancer antigen presentation; STEP3: Priming and activation; STEP4: Trafficking of cells to tumors; STEP5: Infiltration of immune cells into tumors; STEP6: Recognition of cancer cells by T cells; STEP7: Killing of cancer cells. *, $P < 0.05$; **, $P < 0.01$; ***, $P < 0.001$.

Transwell assays further confirmed that silencing SHROOM2 impaired the migratory and invasive abilities of both BC cell lines (Figs. 12A–12B). Consistent with these findings, wound healing assays indicated a reduction in cell migration following SHROOM2 knockdown (Figs. 12C–12D).

## DISCUSSION

This study explored the association between SHROOM2 expression and patient outcomes across multiple cancer types. Additionally, the relationship between SHROOM2 levels and immunomodulators, immune checkpoints, ICI, immune activity, TMB, and drug sensitivity in pan-cancer was evaluated. Finally, the potential oncogenic role of SHROOM2 in BC was validated through *in vitro* experiments.

SHROOM2 expression has been linked to the risk and pathogenesis of esophageal squamous carcinoma, colorectal cancer, and medulloblastoma (*Liu et al., 2024*). One study demonstrated that SHROOM2 inhibited tumor metastasis and EMT by interacting with ROCK (*Yuan et al., 2019*). In the present study, SHROOM2 expression was found to be

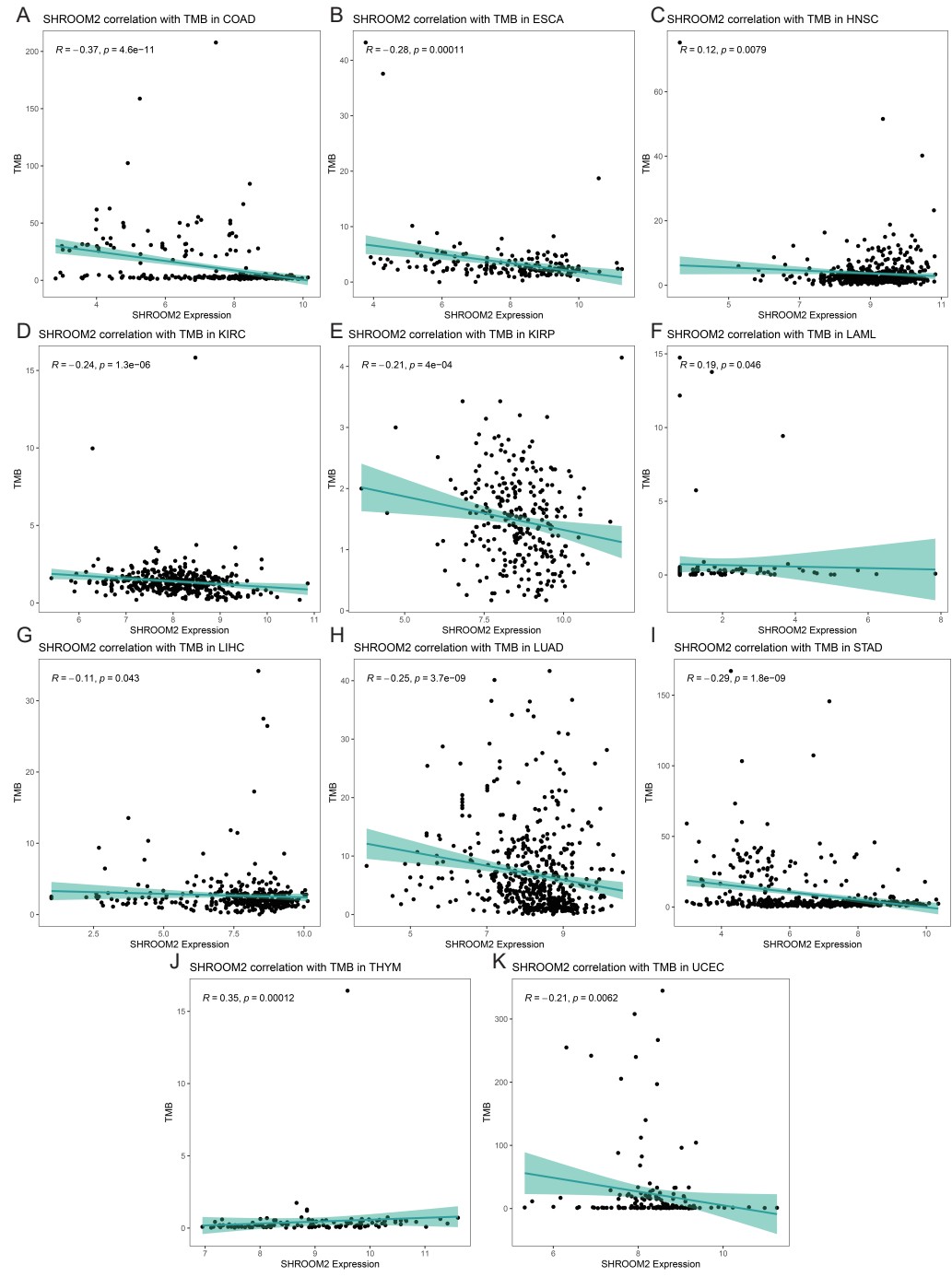

**Figure 9** Correlation between TMB and *SHROOM2* expression in (A) COAD, (B) ESCA, (C) HNSC, (D) KIRC, (E) KIRP, (F) LAML, (G)LIHC, (H) LUAD, (I) STAD, (J) THYM and (K) UCEC.

significantly elevated in various tumor tissues compared to corresponding normal tissues. Survival analysis revealed that SHROOM2 was associated with poor outcomes in BRCA, SARC, and UVM. Furthermore, *in vitro* assays showed that SHROOM2 enhanced the

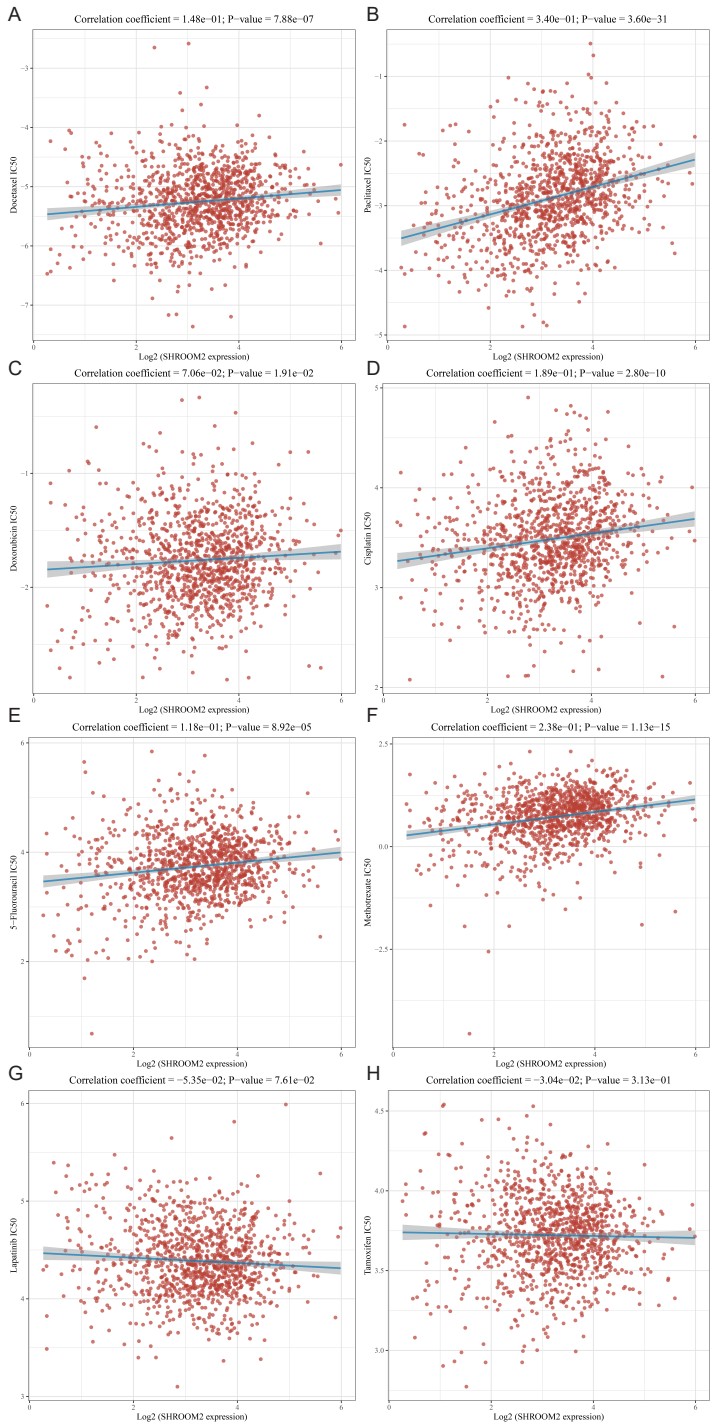

**Figure 10** Correlation between *SHROOM2* expression and the sensitivity of (A) docetaxel, (B) paclitaxel, (C) doxorubicin, (D) cisplatin, (E) 5-fluoruracil, (F) methotrexate, (G) lapatinib and (H) tamoxifen.

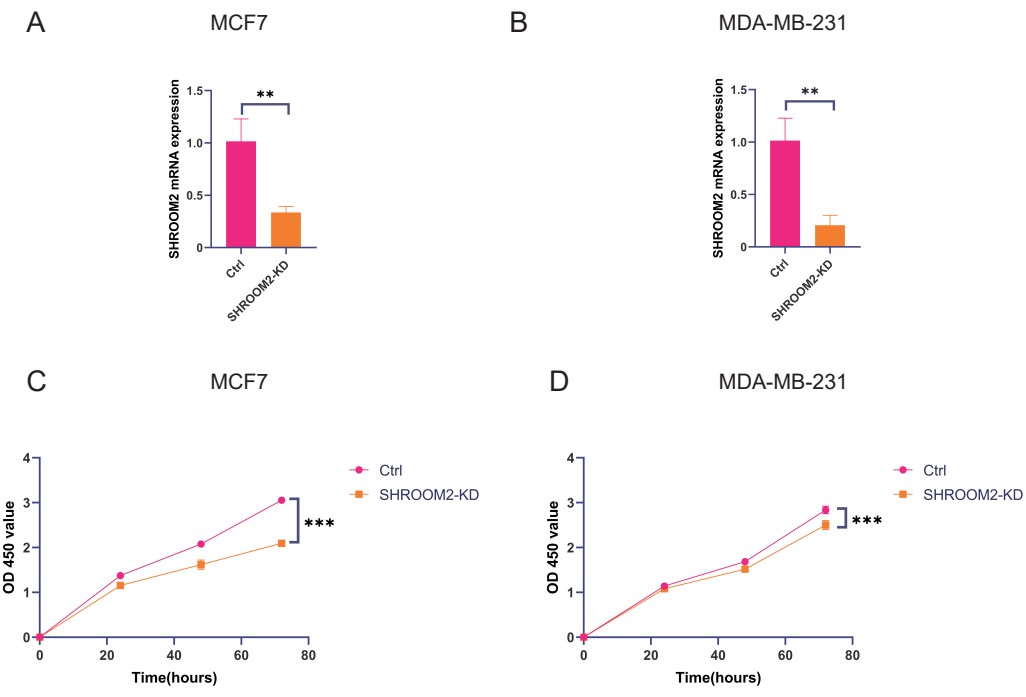

**Figure 11** ***SHROOM2* promoted the proliferative abilities of breast cancer cells *in vitro*.** qRT-PCR validated the knockdown of *SHROOM2* in (A) MCF7 and (B) MDA-MB-231 cells. CCK-8 assay showed that knockdown *SHROOM2* suppressed the proliferation of (C) MCF7 and (D) MDA-MB-231 cells. *, $P < 0.05$; **, $P < 0.01$; ***, $P < 0.001$. CCK-8, Cell Counting Kit-8; qRT-PCR, reverse transcription–quantitative PCR.

migratory, invasive, and proliferative capacities of BC cells. These results suggest that SHROOM2 plays a significant role in the progression of various cancer types.

The tumor microenvironment (TME) plays a critical role in tumor progression, comprising a mesenchymal ecosystem of mesenchyme, immune cells, inflammatory cells, endothelial cells, adipocytes, and fibroblasts (*Ugai et al., 2024*). Immune checkpoints within the TME maintain self-tolerance and regulate inflammation (*Ghaedrahmati, Esmaeil & Abbaspour, 2023*). Tumor cells, however, can exploit immune checkpoint pathways to evade immune-mediated destruction and modulate immune cell function *via* intercellular communication. A study revealed that tumor cells secreted small extracellular vesicles containing cytokines, proteins, and lipids, which facilitated communication between tumor cells and macrophages, driving the polarization of macrophages into tumor-associated macrophages (*Niu et al., 2024*). This phenotypic shift alters the TME's cellular composition and accelerates tumor progression. Immune checkpoint molecules, primarily expressed on immune cells, play a central role in maintaining immune equilibrium. Consequently, therapeutic strategies are being developed to counteract tumor growth by reprogramming the TME. One such strategy, chimeric antigen receptor T-cell immunotherapy (CAR-T) cell therapy, involves the infusion of genetically engineered T cells to remodel the TME. Notably, research indicates that targeting tumor metabolism can enhance CAR-T efficacy

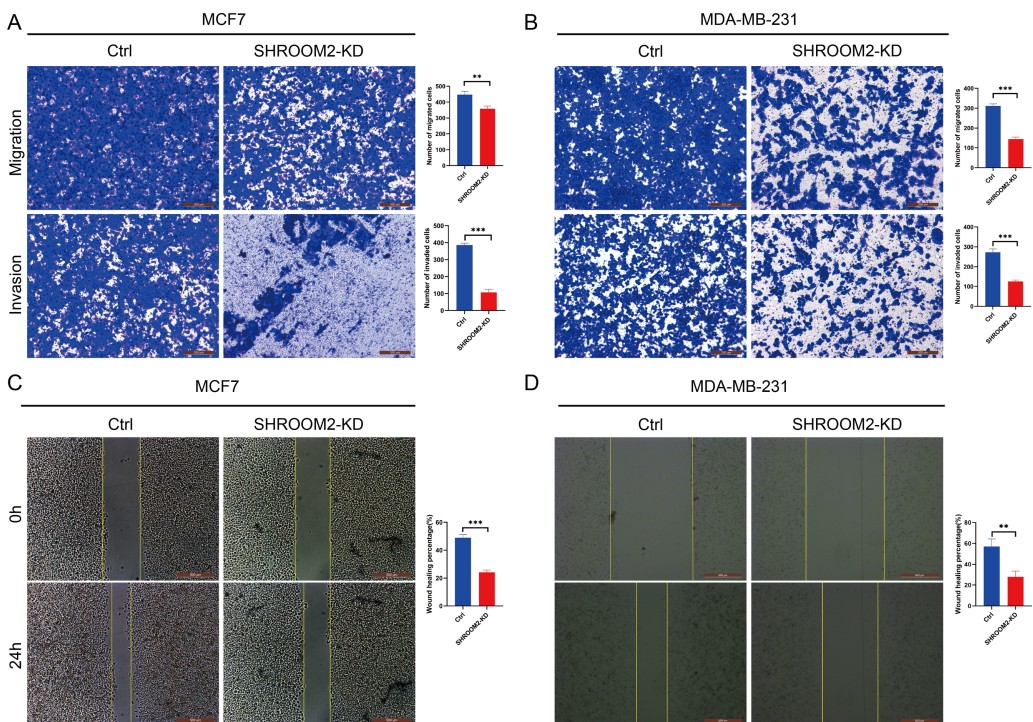

**Figure 12** ***SHROOM2*** **promoted the migratory, and invasive abilities of breast cancer cells** ***in vitro***.
Transwell assays showed that knockdown of *SHROOM2* attenuated the migratory and invasive abilities of
(A) MCF7 and (B) MDA-MB-231 cells. Wound healing assay showed that knockdown of *SHROOM2* sup-
pressed the migration of (C) MCF7 and (D) MDA-MB-231 cells. *, $P < 0.05$; **, $P < 0.01$; ***, $P < 0.001$.

(*Ramapriyan et al., 2024*). Additionally, immune checkpoint inhibitors are a cornerstone of
immunotherapy, directly blocking immune checkpoints to reshape the immunosuppressive
TME (*Kundu et al., 2024*). In particular, PD-1/PD-L1 inhibitors have demonstrated clinical
efficacy in certain malignancies. For example, PD-1 blockade has been shown to significantly
extend progression-free survival in patients with advanced gastric cancer (*Yu et al., 2024*).
However, interpatient heterogeneity in tumor biology leads to variable responses to ICI.
Emerging evidence suggests that liquid biopsy-derived biomarkers, such as circulating PD-
L1 levels, cytokine profiles, and lymphocyte subset distributions, may serve as predictors
of ICI efficacy (*Splendiani et al., 2024*). In this study, SHROOM2 expression was found to
correlate with various immune checkpoint molecules, particularly PD-L1, across multiple
cancer types, suggesting that SHROOM2 may influence immune checkpoint function in
cancer.

In addition to immune checkpoints, immunomodulators play a key role in the TME. The
immune system functions as a complex network designed to protect the body from harmful
agents, eliminate pathogens or cancer cells, maintain memory lymphocytes, and remove
autoreactive immune cells to ensure self-tolerance (*Debele, Yeh & Su, 2020*). Homeostasis
of the immune system depends on both the innate and adaptive immune responses, each
involving distinct cells and molecules that perform specific functions (*Valdés-González*

*et al., 2023*). Cytokines, or immunomodulators, secreted by immune cells in response to specific stimuli, regulate both innate and adaptive immune responses (*Valdés-González et al., 2023*). This study identified a negative correlation between SHROOM2 expression and immunomodulators in BRCA, KIRC, GBM, SARC, THCA, BLCA, and LUSC, and a positive correlation in PRAD, OV, and CESC. These findings indicate that SHROOM2 influences immunoregulatory factors differently across various cancers, potentially through distinct signaling pathways.

TMB quantifies the number of nonsynonymous somatic mutations per megabase (Mb) in a tumor cell genome and is a key predictor of immunotherapy efficacy (*Jardim et al., 2021*). It is typically assessed through whole-exome or targeted panel sequencing. Tumors with elevated TMB accumulate a greater number of mutations, resulting in the production of novel proteins (*Fancello et al., 2019*; *Wang et al., 2024*). These proteins are processed and presented on tumor cell surfaces by major histocompatibility complex (MHC) molecules, forming new antigens. T cells recognize these antigens, triggering immune responses that target and eliminate tumor cells, thereby improving therapeutic outcomes (*Fancello et al., 2019*). In this study, SHROOM2 expression was found to be negatively correlated with TMB across various cancer types, suggesting a potential role of SHROOM2 in modulating immunotherapy efficacy.

Chemotherapy effectively targets and destroys rapidly proliferating tumor cells (*Cui et al., 2020*), while targeted therapies inhibit tumor growth by blocking specific molecular targets such as HER2. This study indicates that SHROOM2 may contribute to chemotherapy resistance in BC.

This study investigated the role of SHROOM2 across various cancers, with a specific focus on BC. Despite utilizing the TCGA database to analyze SHROOM2 expression and its associations with tumor immune features and prognosis in pan-cancer, including BC, several limitations of the database must be acknowledged. The TCGA primarily includes data from primary tumors, excluding cell lines, xenografts, and metastatic specimens. This limitation hinders investigations into tumor evolution, drug resistance mechanisms, and the relevance of preclinical models. For example, while amplifications in glioblastoma are best detected in primary tumors, homozygous deletions are more consistently identified in xenografts and cell lines. Additionally, our study does not investigate the SHROOM2-associated signaling pathways that may modulate immune responses. Future research will involve in-depth *in vitro* and *in vivo* studies to clarify the mechanistic connections between SHROOM2, antitumor immunity, and drug sensitivity.

## CONCLUSIONS

This study demonstrates that SHROOM2 expression is associated with immune-related features, prognosis, and chemotherapy sensitivity across multiple cancer types. These results highlight the relevance of SHROOM2 in pan-cancer malignancies, including BC, and validate its potential as a prognostic biomarker for cancer progression.

## ACKNOWLEDGEMENTS

The authors gratefully acknowledge the technical support provided by The Fourth Hospital of Hebei Medical University.

### Funding

This study was funded by The Open University of China Science Foundation for Youths (Q22I0023). The government-funded Excellent Talents Program of Hebei Provincial Department of Finance (ZF2025187) funded the APC for this article. The funders had no role in study design, data collection and analysis, decision to publish, or preparation of the manuscript.

### Grant Disclosures

The following grant information was disclosed by the authors:
The Open University of China Science Foundation for Youths: Q22I0023.
Government-funded Excellent Talents Program of Hebei Provincial Department of Finance: ZF2025187.

### Competing Interests

The authors declare there are no competing interests. Yuechao Ren is employed by OUC online Education & Information Technology Co., Ltd.

### Author Contributions

- Yaya Wang conceived and designed the experiments, performed the experiments, authored or reviewed drafts of the article, and approved the final draft.
- Yuechao Ren analyzed the data, authored or reviewed drafts of the article, and approved the final draft.
- Xiaoyan Zheng analyzed the data, authored or reviewed drafts of the article, and approved the final draft.
- Yan Wang analyzed the data, prepared figures and/or tables, and approved the final draft.
- Haoqi Wang analyzed the data, prepared figures and/or tables, and approved the final draft.
- Xi Zhang performed the experiments, analyzed the data, prepared figures and/or tables, and approved the final draft.
- Sainan Li conceived and designed the experiments, authored or reviewed drafts of the article, and approved the final draft.

### Data Availability

  The data is available in the Supplementary Files.

## Supplemental Information

Supplemental information for this article can be found online at http://dx.doi.org/10.7717/peerj.20051#supplemental-information.

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
