# Peer review of "Identification of immunological and prognostic value of SHROOM2 in pan-cancer and experimental verification of its role in promoting malignant phenotypes in breast cancer"

_PeerJ, doi:10.7717/peerj.20051_

## Round 0.1 · original submission · Major Revisions

**Language Note:** When you prepare your next revision, please either (i) have a colleague who is proficient in English and familiar with the subject matter review your manuscript, or (ii) contact a professional editing service to review your manuscript. PeerJ can provide language editing services - you can contact us at [email protected] for pricing (be sure to provide your manuscript number and title). – PeerJ Staff

Reviewer 1 ·

Basic reporting

This is a well-rounded study that combines bioinformatics and experimental validation to explore the role of SHROOM2 in pan-cancer, with a particular focus on breast cancer. The authors analyzed large datasets (TCGA, GTEx), performed immune correlation studies, explored links with TMB and drug sensitivity, and validated SHROOM2's function using knockdown experiments in cell lines. That’s quite comprehensive, and the idea is timely. I think the manuscript has good potential, but it needs polishing before publication.

Specific comments

Introduction:
To underscore the critical need for continued oncology research, the manuscript should incorporate up-to-date figures on overall cancer incidence alongside the prevalence and survival rates specific to the cancer type under investigation (Cancer Statistics, 2024). A succinct overview of the evolution of cancer therapeutics, anchored in the NIH’s comprehensive review “Cancer treatments: Past, present, and future, 2024,” will contextualize current modalities within their historical development. Moreover, integrating a comparative discussion of treatment paradigms as detailed in “Different strategies for cancer treatment: Targeting cancer cells or their neighbors? “(2025) will clarify the relative merits of directly attacking malignant cells versus modulating their microenvironment. Finally, framing a section on emerging directions in oncology by drawing on “Overview of perspectives on cancer, newer therapies, and future directions” will provide readers with a forward-looking roadmap for novel therapeutic strategies.

Improve flow and avoid repetition; phrases like “SHROOM2 level was linked to…” are repeated too many times. Try varying your wording and focusing more on the biological meaning of these findings.

The manuscript depth could be further enriched by detailing some of the analytical approaches, particularly the use of generative adversarial networks (GANs) in transcriptomic studies. For instance, incorporating a discussion of “Generative adversarial networks applied to gene expression analysis: An interdisciplinary perspective” would highlight how GANs can synthesize realistic gene expression profiles, augment limited datasets, and uncover novel regulatory patterns. This addition would not only contextualize your computational workflow but also underscore the innovative methodologies driving discovery in high-dimensional genomic analysis of SHROOM2 expression.

Expand the discussion:
Bring in more interpretation of how SHROOM2 could be used clinically. Could it serve as a biomarker for immunotherapy response? The authors should expand their discussion of contemporary cancer immunotherapy to encompass advances in checkpoint inhibition, cell engineering, and microenvironmental modulation. For instance, in melanoma, circulating biomarkers such as soluble PD-L1, cytokine profiles, and lymphocyte subsets have shown promise in predicting response to PD-1/PD-L1 blockade (“Immunotherapy in melanoma: Can we predict response to treatment with circulating biomarkers?”, 2024). Meanwhile, next-generation CAR T-cell strategies that adapt to altered tumor metabolism, by enhancing mitochondrial fitness or resisting high lactate levels, are being explored to overcome the metabolic barriers of solid tumors (“Altered cancer metabolism and implications for next-generation CAR T-cell therapies”, 2024). In gastric cancer, evolving insights into the PD-1/PD-L1 axis underscore the potential of combinatorial approaches targeting additional checkpoints and tailoring treatment based on tumor microenvironment profiling (“Evolving perspectives regarding the role of the PD-1/PD-L1 pathway in gastric cancer immunotherapy”, 2025). Finally, interrupting the crosstalk between tumor cells and tumor-associated macrophages mediated by small extracellular vesicles offers a novel avenue to reprogram immunosuppressive macrophages and bolster anti-tumor immunity (“Small extracellular vesicles-mediated cellular interactions between tumor cells and tumor-associated macrophages: Implication for immunotherapy”, 2025).

Is there potential for targeting it therapeutically? Discuss the bias from TCGA.

Experimental design

-

Validity of the findings

-

Reviewer 2 ·

Basic reporting

-

Experimental design

-

Validity of the findings

-

Additional comments

The authors investigate the immunological and prognostic significance of SHROOM2 across multiple cancer types and experimentally validate its role in promoting malignant behavior in breast cancer (BC). Specifically, they ask whether SHROOM2 expression:
• Is differentially regulated in tumor versus normal tissues across TCGA and GTEx datasets,
• Correlates with patient survival, tumor mutation burden (TMB), immune‐related features (immunomodulators, checkpoints, immune cell infiltration), and drug sensitivity, and
• Functionally contributes to the proliferation, migration, and invasion of BC cells in vitro.

This question is important because SHROOM2, a member of the Shroom family implicated in morphogenesis and previously linked to several cancer types, has not been comprehensively analyzed in a pan‐cancer context nor mechanistically studied in breast cancer. Understanding its role could identify a novel biomarker and therapeutic target in BC and other malignancies. To address this, the authors combine bioinformatic analyses of RNA‐seq and clinical data from TCGA and GTEx (differential expression, survival Cox models, GSEA/GO/KEGG, Spearman correlations with immune features via TISIDB, TIMER2.0, ESTIMATE, TMB via maftools, drug response via pRRophetic) with in vitro experiments (siRNA knockdown of SHROOM2 in MCF7 and MDA‐MB‐231 cells, followed by qRT‐PCR, CCK‐8 assays, transwell migration/invasion, and wound‐healing assays).

They conclude that SHROOM2 is broadly upregulated in tumors, associated with worse survival in several cancers (including BC), correlates with immunomodulatory pathways, TMB, and chemotherapeutic resistance, and that its depletion in BC cells reduces proliferation, migration, and invasion. Thus, SHROOM2 may serve as an immunological and prognostic biomarker and a potential therapeutic target in breast cancer and other malignancies.
Overall, this study is well structured. I just have some minor comments for the authors to improve their manuscript.

1. Literature references and background: The Introduction briefly surveys SHROOM2 roles in other cancers but expands on the specific knowledge gap in breast cancer.

2. Figure 6A is too blurry, and I can’t read anything.

3. Some descriptions (e.g., qRT‐PCR steps, reagent catalog numbers) are overly verbose; critical details (e.g., siRNA transfection protocol, cell culture conditions, number of biological replicates) should be consolidated into a clear, reproducible Methods table.

4. Overall, the conclusions are well supported by results. However, statements implying SHROOM2 as a therapeutic target should be tempered until in vivo or mechanistic validations are available (e.g., interaction with PD-L1 pathways).

Reviewer 3 ·

Basic reporting

Wang et al. attempted in the submitted manuscript to establish a comprehensive understanding of SHROOM2 in pan-cancer with a focus on breast cancer, specifically. The article was written in professional English, and the reviewer has no concern about the grammar of the text.

The topic was poorly motivated in the Introduction section, with multiple redundancies in the first two paragraphs and a lack of sufficient background on SHROOM2. Line 47-18 introduced that about 287,850 cases of breast cancer occur in American women, which is unrelated to the prior and following contents. Are the authors trying to stress the high number of cases of BC? In which case, the authors should also explain if this number ranks high in the overall cancer cases. Or are the authors trying to establish that American women are at high risk? Then, a breakdown of the number of cases by geographic information should be supplied. Moreover, Lines 52-59 are also completely out of place. If the authors were to give a comprehensive review of the Shroom protein family, then they should introduce each one of them sequentially and in detail. For example, the authors mentioned that Shroom 2-4 has a protein domain and that SHROOM3 does not function on this domain. How is this domain related to the functions of Shrooms in cancer, and why does it matter that it is not functional in SHROOM3 specifically? Does SHROOM1 have this domain at all? How do each of these Shroom subtypes correlate and differ from each other, and what makes SHROOM2 unique to be the topic of this manuscript?

Besides, the title of the article specifically mentions the immunological value of SHROOM2, but this topic is not motivated at all in the Introduction. Does SHROOM2 play a known role in the immune system that made the authors investigate its value in cancers? Such motivation is lacking.

Experimental design

The research question is ambiguously defined. The authors seem to want to understand the role of SHROOM in all cancer types and used breast cancer as an example. First, it is unclear throughout the texts if the authors wanted to perform a pan-cancer meta-analysis or if they truly wanted to focus on breast cancer. Second, based i Figure 1A, it appears that breast cancer is not the top cancer type that has an elevated SHROOM2 expression. Why choose BC instead of, say, SKCM, based on this figure? Last but not least, the authors started the article by introducing how aggressive breast cancer is, but jumped between the pan-cancer concept and breast cancer in the Results. Therefore, the reviewer considers the research question to be poorly defined, as it is unclear what exactly the authors want to convey.

A minor issue with the methods is a lack of details on survival analysis. The authors should explain which survival analysis method was used and what each of the "overall survival", "disease-specific survival", and "progression-free interval" means. Methods should be described with sufficient details, but none are included in the current texts.

Validity of the findings

Overall, the findings are not well supported by the methods and data. For example, in Results under the section "SHROOM2 was correlated with multiple BC-related pathways," Figure 4D suggests that "nicotine addiction pathways" and "neuroactive ligand-receptor interactions" from KEGG were enriched. How are these two pathways related to BC? Is there any literature that suggests that? How do we know if any of the pathways shown in the manuscript are BC-specific, anyway?

In Figure 1D-F, the authors copied and pasted whatever was shown on GeneCard as part of the "data". However, no analysis was done here with this public information, as opposed to what the authors claimed in Lines 233-234, "we analyzed...". What kind of conclusion pertaining to the research topic can the authors draw from these figures?

Additional comments

The authors should significantly increase the font sizes of Figures 4-6, as these three figures are currently completely illegible.

---

## Round 0.2 · accepted · Accept

All concerns of the reviewers were adequately addressed, and the revised manuscript is acceptable now.

Reviewer 1 ·

Basic reporting

-

Experimental design

ok

Validity of the findings

-

Additional comments

The paper has been improved

Reviewer 2 ·

Basic reporting

The revised manuscript is clear with professional English, the references are correct and up-to-date, and the figures are clear.

Experimental design

The revised experimental design is proper, and the research question is well defined.

Validity of the findings

The revised manuscript showed robust and statistically sound findings. The conclusions are well stated.

Reviewer 3 ·

Basic reporting

The authors have addressed the comments the reviewer made in the previous round of the review. No further comment.

Experimental design

-

Validity of the findings

-